# Risk of Vestibulocochlear Disorders in Patients with Migraine or Non-Migraine Headache

**DOI:** 10.3390/jpm11121331

**Published:** 2021-12-08

**Authors:** Sang-Hwa Lee, Jong-Ho Kim, Young-Suk Kwon, Jae-June Lee, Jong-Hee Sohn

**Affiliations:** 1Department of Neurology, Chuncheon Sacred Heart Hospital, Hallym University College of Medicine, Chuncheon 24253, Korea; neurolsh@hallym.or.kr; 2Institute of New Frontier Research Team, College of Medicine, Hallym University College of Medicine, Chuncheon 24253, Korea; poik99@hallym.or.kr (J.-H.K.); gettys@hallym.or.kr (Y.-S.K.); iloveu59@hallym.or.kr (J.-J.L.); 3Department of Anesthesiology and Pain Medicine, Chuncheon Sacred Heart Hospital, Hallym University College of Medicine, Chuncheon 24253, Korea

**Keywords:** migraine, headache, vestibular disorder, cochlear disorder, tinnitus, hearing loss

## Abstract

Headaches, especially migraines, have been associated with various vestibular symptoms and syndromes. Tinnitus and hearing loss have also been reported to be more prevalent among migraineurs. However, whether headaches, including migraine or non-migraine headaches (nMH), are associated with vestibular and cochlear disorders remains unclear. Thus, we sought to investigate possible associations between headache and vestibulocochlear disorders. We analyzed 10 years of data from the Smart Clinical Data Warehouse. In patients with migraines and nMH, meniere’s disease (MD), BPPV, vestibular neuronitis (VN) and cochlear disorders, such as sensorineural hearing loss (SNHL) and tinnitus, were collected and compared to clinical data from controls who had health check-ups without headache. Participants included 15,128 with migraines, 76,773 patients with nMH and controls were identified based on propensity score matching (PSM). After PSM, the odds ratios (OR) in subjects with migraine versus controls were 2.59 for MD, 2.05 for BPPV, 2.98 for VN, 1.74 for SNHL, and 1.97 for tinnitus, respectively (*p* < 0.001). The OR for MD (1.77), BPPV (1.73), VN (2.05), SNHL (1.40), and tinnitus (1.70) in patients with nMH was also high after matching (*p* < 0.001). Our findings suggest that migraines and nMH are associated with an increased risk of cochlear disorders in addition to vestibular disorders.

## 1. Introduction

Dizziness or vertigo is a relatively common symptom in headache patients [1]. Reportedly, more than half of headache patients experience dizziness accompanied by headache. In particular, vestibular symptoms accompanying headache are very common in migraineurs, and the lifetime prevalence of vertigo or dizziness in migraineurs is significantly higher than in controls [2,3]. In addition, in various studies on migraine and tension-type headache (TTH) patients, the frequency of vestibular symptoms such as dizziness, vertigo, imbalance, postural instability, and vestibular disorders was high, and abnormalities in vestibular function tests have been reported [4,5,6,7,8,9]. Based on the relationship between migraines and vestibular symptoms reported in numerous studies, vestibular migraine (VM) was included in the appendix of the 3rd edition of the International Classification of Headache Disorders. VM is closely associated with Meniere’s disease (MD), and a substantial overlap between VM and MD has been reported [10,11,12,13,14]. In a previous study, patients presented with simultaneous VM and MD signs/symptoms and a new clinical syndrome, VM/MD overlap syndrome, was proposed [15]. Other authors reported a number of patients who experienced migraine-related symptoms and family histories presenting with long-term, unilateral, fluctuating hearing loss with aural fullness and tinnitus who never develop vertigo or only mild dizziness that did not meet the strict criteria for VM. Therefore, the authors proposed a new concept of cochlear migraine for this condition [16]. Although the pathophysiology of migraine including VM is not well known, the most accepted theories to date are thought to be vascular mechanisms and/or inflammation and neurotransmitter mechanisms. In addition, migraines reportedly damage the cochlea of the inner ear via a neurovascular mechanism [17]. In previous population-based cohort studies, migraines were suggested to increase the risk of tinnitus, sensorineural hearing impairment, and/or sudden deafness [18,19]. In addition, the relationship between headaches and tinnitus or hearing impairment has been described in several clinical studies [20,21,22,23,24,25]. In cross-sectional studies, several types of headaches, including migraine and TTH, were shown to be associated with tinnitus and hearing impairment [22,25]. In a population-based study, the risk of tinnitus, sensorineural hearing impairment, and sudden deafness were found to be significantly higher in patients with non-migraine headache than in subjects without headache [20]. In other studies, tinnitus was found to be associated with temporomandibular joint and/or cervical dysfunction, or related to pericranial muscle tenderness in patients with headache and craniofacial pain [21,23,24]. 

However, whether headaches, including migraine and non-migraine headaches, are associated with cochlear disorders such as tinnitus or hearing loss in addition to vestibular disorders remains unclear. Thus, the associations between headaches, including migraine and non-migraine headaches, and vestibulocochlear disorders were investigated in the present study using the Smart Clinical Data Warehouse (CDW) over a period of 10 years. 

## 2. Materials and Methods

### 2.1. Subjects

We retrospectively analyzed clinical data from the Smart CDW of Hallym University Medical Center (HUMC) from January 2011–April 2021. The Smart CDW, based on the QlikView Elite Solution (Qlik, Lund, Sweden), is used at the four hospitals of the HUMC and offers electronic medical record text data analysis and an integrated analysis of fixed data. Patients eligible for the study included those with migraines, 20–80 years of age, diagnosis of migraine by a board-certificated neurologist, and ≥2 consecutive visits to the neurology department. Patients with non-migraine headache, 20–80 years of age and having a diagnosis determined by the presence of two codes were also eligible. Patients with headaches or TTH and subjects with a history of more than one visit for migraines were excluded. The information on comorbidities and migraine medications, such as non-steroid anti-inflammatory drugs (NSAIDs) and triptans, were collected for analysis. Triptan drugs are specifically used to treat migraine; those used at HUMC include almotriptan, frovatriptan, naratriptan sumatriptan, and zolmitriptan. The control group included patients 20–80 years of age who had undergone general health checkups at a health promotion center. Patients with a history of headache or migraine assessed using a basic questionnaire completed prior to the health examination and patients with a history of headache or migraine who visited our medical center were excluded. This study was approved by the Clinical Research Ethics Committee of Chuncheon Sacred Heart Hospital, Hallym University (IRB No. 2021-06-003). Since only deidentified data were used in this study, the review board waived the requirement for informed consent from all subjects.

### 2.2. Migraine/Non-Migraine Headache, Vestibulocochlear Disorders, and Covariates 

Among the enrolled groups, the medical records of subjects with a diagnosis of MD, benign paroxysmal positional vertigo (BPPV), vestibular neuronitis (VN), or cochlear disorders such as sensorineural hearing loss (SNHL) and tinnitus, were collected and compared with clinical data from controls without headache who had health check-ups. In addition, the presence of comorbidities was defined according to the presence of International Classification of Diseases, tenth revision (ICD-10) codes in the database and included sleep disorders (ICD-10 codes F51, G258, G47), heart disease (ICD-10 codes I05–09, I21–23, I30–47, I49–52), hypertension (ICD-10 codes I10–15), diabetes mellitus (ICD-10 codes E10–14), dyslipidemia (ICD-10 code E78), renal failure (ICD-10 codes N03, N18–17), chronic hepatitis (ICD-10 codes B18, I85, K70–74), anxiety disorder (ICD-10 code F41), depression (ICD-10 codes F31–34, F412, F432), cerebrovascular diseases (ICD-10 codes G45–46, I60–69), menopause (ICD-10 codes M800, M010, N924, N95), chronic pulmonary disease (ICD-10 codes J40–47), angina (ICD-10 codes I20, I24, I251), and atrial fibrillation (ICD-10 codes I480–482, I489).

### 2.3. Statistical Analysis

Continuous data are presented as means and standard deviations (SDs), and categorical data as frequencies and percentages. A *t*-test was performed to compare the continuous data of patients with migraine or non-migraine headache. Categorical data were analyzed using the chi-square test. First, odds ratios (ORs) with 95% confidence intervals (CIs) were calculated for the occurrence of MD, BPPV, VN, SNHL, and tinnitus for each variable using logistic regression. The OR is a measure of the association between an exposure and outcome and, in this study, represents the likelihood of MD, BPPV, VN, SNHL, and tinnitus given a particular exposure compared with the likelihood in its absence. Fully adjusted ORs for MD, BPPV, VN, SNHL, and tinnitus were then calculated for each variable including migraine and non-migraine headaches. 

Since patients could not be randomized based on the presence of migraine, confounding and selection biases were accounted for using propensity scores. In this study, propensity score matching (PSM) was conducted for normal controls and subjects with migraine or non-migraine headache. Python (version 3.7; Anaconda Inc., Austin, TX, USA) and Pymatch (version 0.3.4; https://github.com/benmiroglio/pymatch accessed on 20 October 2021) were used for PSM. The propensity scores ranged from 0.07–0.87. All matched cases had scores within 0.0001 of each other and the matching ratio was 1:1 (15,128 migraine patients and 15,128 normal controls; 14,606 non-migraine headache patients and 14,606 normal controls). 

The ORs for MD, BPPV, VN, SNHL, and tinnitus in migraine and non-migraine headache patients were compared with the normal controls. In the analysis, covariates and propensity scores were used to calculate adjusted ORs. All *p*-values were two-sided and a *p*-value < 0.05 was considered statistically significant. SPSS software (version 24.0; IBM Corp., Armonk, NY, USA) was used for statistical analyses. 

## 3. Results

### 3.1. Subject Characteristics 

The present study included 15,128 patients with migraines and 76,773 patients with non-migraine headaches identified between January 2011 and April 2021. A total of 315,330 subjects without migraines or headaches were used as controls and selected from the same database. The enrollment process is presented in Figure 1. Among the 15,128 patients with migraine and 76,773 patients with non-migraine headache, 11,247 (74.3%) and 45,148 (58.8%) were female, and the mean (SD) age was 44.0 (14.3) and 49.1 (14.6) years, respectively. In addition, 15,128 and 76,773 controls for migraine and non-migraine headache, respectively, were identified based on PSM (Figure 1).

Between the migraine and control groups, all the absolute standardized differences were <0.1 after PSM (Table 1). In addition, significant differences were not observed for any variable after PSM between non-migraine headache and control groups (Table 2). In subgroup analysis, significant difference was not observed between the migraine group taking triptan medication and controls or between the migraine group taking NSAID medication and controls, for any variable after PSM.

### 3.2. ORs for Vestibulocochlear Disorders in Migraineurs 

Before PSM, the unadjusted ORs in subjects with migraine versus controls were 2.519 (95% CI, 1.988–3.192; *p* < 0.001) for MD, 1.941 (95% CI, 1.728–2.182; *p* < 0.001) for BPPV, 2.838 (95% CI, 2.518–3.200; *p* < 0.001) for VN, 1.647 (95% CI, 1.331–2.038; *p* < 0.001) for SNHL, and 1.869 (95% CI, 1.574–2.215; *p* < 0.001) for tinnitus. The ORs for MD (OR, 2.597; 95% CI, 2.047–3.295; *p* < 0.001), BPPV (OR, 2.045; 95% CI, 1.816–2.302; *p* < 0.001), VN (OR, 2.976; 95% CI, 2.636–3.360; *p* < 0.001), SNHL (OR, 1.739; 95% CI, 1.404–2.156; *p* < 0.001), and tinnitus (OR, 1.970; 95% CI, 1.658–2.341; *p* < 0.001) in patients with migraine were also high after PSM (Table 3). Subgroup analysis showed the ORs in the migraine group taking triptans were 2.928 (95% CI, 2.038–4.206; *p* < 0.001) for MD, 2.359 (95% CI, 1.960–2.838; *p* < 0.001), 2.882 (95% CI, 2.385–3.483; *p* < 0.001), for 1.812 (95% CI, 1.291–2.543; *p* < 0.001) for SNHL, and 1.862 (95% CI, 1.446–2.399; *p* < 0.001) for tinnitus after PSM. The ORs for MD (OR, 1.809; 95% CI, 1.360–2.408; *p* < 0.001), BPPV (OR, 2.433; 95% CI, 2.095–2.826; *p* < 0.001), VN (OR, 3.365; 95% CI, 2.887–3.921; *p* < 0.001), SNHL (OR, 1.649; 95% CI, 1.274–2.133; *p* < 0.001), and tinnitus (OR, 1.941; 95% CI, 1.566–2.405; *p* < 0.001) in the migraine group using NSAIDs were also high after PSM (Table 4).

### 3.3. ORs for Vestibulocochlear Disorders in Non-Migraine Headaches 

Table 5 shows the ORs for development of vestibular and cochlear disorders in non-migraine headache patients. The unadjusted and adjusted ORs for MD, BPPV, VN, SNHL, and tinnitus were lower in non-migraine headache patients than in migraine patients. After PSM, the adjusted ORs in subjects with non-migraine headache versus controls were 1.771 (95% CI, 1.560–2.011; *p* < 0.001) for MD, 1.731 (95% CI, 1.637–1.831; *p* < 0.001) for BPPV, 2.048 (95% CI, 1.935–2.168; *p* < 0.001) for VN, 1.396 (95% CI, 1.273–1.531; *p* < 0.001) for SNHL, and 1.693 (95% CI, 1.569–1.826; *p* < 0.001) for tinnitus (Table 5).

## 4. Discussion

In the present study, the risk of cochlear disorders in patients with migraine or non-migraine headaches, in addition to vestibular disorders, was investigated based on data from the Smart CDW of HUMC over a 10-year period. The study included 15,128 patients with migraines, 76,773 patients with non-migraine headache, and 315,330 non-headache controls. After PSM, patients with migraine or non-migraine headache were at significantly greater risk of MD, BPPV, VN, SNHL, and tinnitus than control subjects without headache. 

Migraines are associated with various vestibular symptoms and several vestibular syndromes [26]. In addition, several vestibular laboratory abnormalities in migraineurs have been identified in previous studies [27]. VM is a common cause of recurrent episodic vertigo, accompanied by migraine-related symptoms [28]. VM is closely associated with MD and a significant overlap between these two diseases has been reported [10,11,12,13,14]. In a previous report, patients presenting simultaneously with VM and MD sign/symptoms was proposed as a new clinical syndrome and termed VM/MD overlap syndrome [15]. In other studies, a number of patients who experienced migraine-related symptoms and family histories presenting with long-term, unilateral, fluctuating hearing loss with aural fullness and tinnitus (but without vertigo or only mild dizziness) that did not meet the strict criteria for VM were reported. Therefore, the authors proposed a new concept of a cochlear migraine [16]. In previous epidemiological studies, vestibulocochlear signs and symptoms were shown to be common in patients with migraine. Similar to the results of previous studies, our results showed that the ORs for vestibulocochlear disorders, including MD, BPPV, VN, SNHL, and tinnitus, were significantly higher in migraine patients compared with controls. The non-migraine headache group also showed similar results to the migraine group. The ORs showed a significantly greater risk of vestibular and cochlear disorders in non-migraine headache patients (although each ORs was lower in non-migraine headache patients than in migraine patients). The non-migraine cohort in the present study included patients with codes having TTH or non-specific headache and excluded all types of migraines. Thus, the non-migraine headache cohort mainly included patients with TTH.

In several studies targeting patients with TTH, the frequency of vestibular signs/symptoms was high, and abnormalities were reported in vestibular function tests [7,9]. Furthermore, in several cross-sectional studies, various types of headaches were shown to be associated with tinnitus and hearing impairment including TTH, mixed headache, and headaches combined with cervical muscle tenderness [22,24,25]. In population-based cohort studies from Taiwan, a relationship was observed between primary headaches (mainly in patients with TTH and cochlear disorders) [20,29]. In one study, the non-migraine headache cohort was associated with the risk of tinnitus, sensorineural hearing impairment, and sudden deafness [20], whereas in another study, an association was observed between TTH and sudden sensorineural hearing loss [29]. In particular, tinnitus, defined as the perception of sound in absence of a corresponding sound source, can be present in patients with or without hearing loss and its cause may be multifactorial. In previous studies, the specific subtypes of tinnitus, such as somatosensory and cervicogenic somatic tinnitus, were shown to be associated with increased pericranial and cervical muscle tenderness in patients with migraines, TTH, myogenous pain, and arthrogenous temporomandibular joint disorders [21,22,23,24]. The authors suggested that tinnitus was associated with increased pericranial and cervical muscle tenderness and not any particular form of headache in patients with headache and craniofacial pain [24]. In patients with migraines and TTH, both episodic and chronic, are combined with pericranial and cervical muscle tenderness. In particular, prior study results support the pathogenetic importance of muscular and stress factors in TTH [30,31,32,33,34,35]. In another study, headache laterality was significantly associated with tinnitus laterality, and in the majority of patients, fluctuations in symptom severity of tinnitus and headache were interrelated [35]. Thus, tinnitus and TTH may have common pathophysiological mechanisms. Vascular insufficiency has been proposed as the main pathogenesis of SNHL since many vascular events are associated with SNHL. In a prior study, the sympathetic response to stress was hypothesized to cause vasoconstriction, hemoconcentration, and microcirculation occlusion in the inner ear, which could occur afterwards, ultimately leading to SNHL [36]. In another cohort study, acutely increased neutrophil counts and IL-6 level and a decrease in natural killer cell activity were associated with severe hearing loss and poor prognosis. These results indicate that inflammatory biomarkers and stress are involved in the pathophysiology of SNHL [37]. Thus, the relationship between SNHL and TTH could share pathophysiological mechanisms due to stress-induced inflammatory activities. 

In addition, the ORs for each cochlear disorder in migraine and non-migraine headache patients were lower than for vestibular disorders in the present study. The ORs for each vestibulocochlear disorder were higher in migraine than in non-migraine headache patients. In most prior studies, the association between migraine or each headache group and vestibular or cochlear disorder was investigated; the association between various headache groups and vestibulocochlear disorders have been simultaneously compared in only a few studies. In a large cross-sectional study among students, the adjusted ORs of tinnitus were 1.77 for migraine and 1.38 for non-migraine headache, somewhat lower than in the present study. In the previous study, stronger association was found for migraine with aura and no significant association for non-migraine headache after adjustment [38]. Furthermore, vestibular symptoms were shown to be more common in subjects considered to have definite migraine than in subjects considered to have definite TTH [2,39]. Even in younger patients presenting with a primary headache disorder, the presence of vestibular symptoms helps differentiate migraine from TTH [2]. To date, the association between migraine and vestibular disorder has been generally established. Therefore, VM has been included in the appendix criteria of the International Classification of Headache Disorders-3. More recently, since reports of the association between cochlear disorder and migraine are increasing, a new concept of “cochlear migraine” or “vestibulocochlear migraine” should be considered. In addition, associations between non-migraine headaches and vestibular or cochlear disorders have been reported. Thus, additional studies on the association with non-migraine headaches are needed. 

Vestibulocochlear signs and symptoms have been reported to be common in patients with migraines as well as with other types of primary headache. However, the pathogenesis of these signs and symptoms is unclear. In previous studies, multiple potential interactions between the trigeminal and vestibular systems at various levels were reported. In migraine patients, stimulation of the trigeminal nuclei produced spontaneous nystagmus [40]. Conversely, vestibular nuclei receive both serotonergic inputs from the dorsal raphe nucleus and noradrenergic inputs from the locus coeruleus, and activation of these pain structures during migraine can affect central vestibular processing [27]. These reciprocal connections between the vestibular nuclei and trigeminal nucleus caudalis may include a mechanism in which vestibular signals influence the trigeminovascular pathway and trigeminal information processing during migraine attacks [41]. Furthermore, vestibular and cranial nociceptive pathways possess similarities in neurochemical milieu and express serotonin receptor subtypes that are targets for anti-migraine drugs, such as triptans. The serotonergic 5-HT1B and 5-HT1D receptors are expressed prominently in the inner ear of rats and monkeys [42]. All vestibular and considerable numbers of trigeminal ganglia show positive 5-HT1F receptor immunoreactivity in monkeys. In addition, similar findings in cochlear inner hair cell afferents were observed and these results are applicable to migraine-related tinnitus [43]. In other previous experimental studies, the trigeminal ganglion was shown to innervate the cochlear nucleus and superior olivary complex in guinea pigs. This projection may be associated with integration mechanisms involving the auditory end organ and its central targets [44]. In clinical studies, histopathological findings indicate that sudden deafness in migraine patients resulted from ischemia [17], and ischemia was most likely to be due to migraine-associated vasospasm of the cochlear vasculature [17,45]. In the electrophysiologic study for monitoring cochlear status, subclinical dysfunction of cochlear efferents was observed in migraine patients [46]. Additional experimental studies are needed to elucidate the various potential interactions between the trigeminal nociceptive pathway and vestibular or cochlear systems.

The present study had several limitations. The study was retrospective in design; thus, unmeasured confounders may not have been accounted for. In addition, data were used from subjects who visited one university medical center with four hospitals. Therefore, generalizing the results to the general population is difficult and the possibility of selection bias must be considered. In addition, detailed clinical information regarding headache characteristics or vestibulocochlear disorders was not collected. The symptoms of VM may mimic to the those of MD and BPPV [47]. In particular, the primary differential diagnosis is MD. The current diagnostic criteria for both of these disorders are mainly based on the clinical symptoms of a patient since no biological markers are available. Also, the signs and symptoms of VM sometimes overlap with those of MD, which may give rise to diagnostic uncertainty [10,11,12,13,14,48]. In the early stages of these disorders, a differential diagnosis between MD and VM may be difficult. Due to the similar clinical presentations of these disorders, a follow-up assessment is the only method that can accurately distinguish between VM and MD since the most reliable distinguishing features of patients with MD are progressive hearing loss that manifests over several years and low-frequency hearing loss evident on an audiogram [49]. Therefore, the diagnosis of vestibulocochlear disorders may be difficult and uncertain, which can reduce the accuracy of diagnostic coding in these cases. To evaluate the accuracy of diagnosing the vestibulocochlear disorders, longitudinal follow-up clinical investigation, including detailed clinical features and laboratory test for vestibulocochlear function are required. Furthermore, future prospective, population-based studies are needed to investigate further the association between headache and vestibulocochlear disorders.

## 5. Conclusions

The results of the present study indicate that patients with migraines and non-migraine headaches are at an increased risk of cochlear disorders, such as SNHL and tinnitus, in addition to vestibular disorders. Furthermore, the ORs for each vestibulocochlear disorder were higher in migraine than in non-migraine headache patients. Thus, we propose the concept of “cochlear migraine” or “vestibulocochlear migraine”. Additional experimental studies to elucidate the potential interactions between the trigeminal nociceptive pathway and vestibular or cochlear systems and prospective, population-based studies to investigate the association between headache and vestibulocochlear disorders are needed.

## Figures and Tables

**Figure 1 jpm-11-01331-f001:**
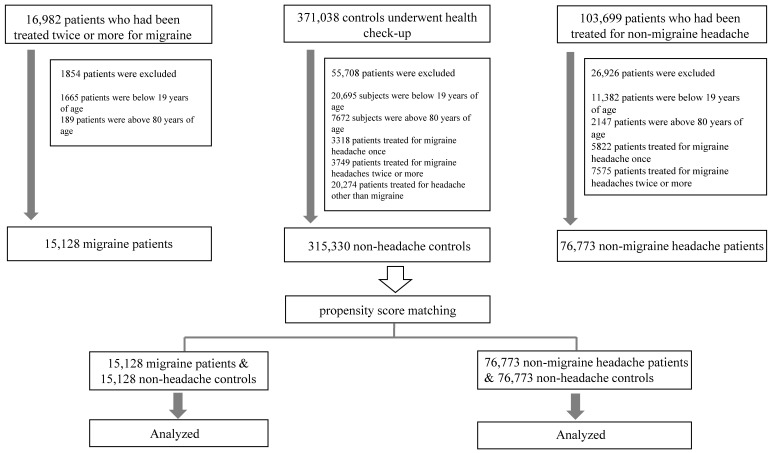
Flow chart of the enrollment process.

**Table 1 jpm-11-01331-t001:** Characteristics of the migraine and control groups before and after PSM.

	Before PSM			After PSM		
	Migraine (*n* = 15,128)	Control (*n* = 315,330)	ASD	Migraine (*n* = 15,128)	Control (*n* = 15,128)	ASD
Age, years (min, SD)	44.0 (14.3)	46.6 (15.4)	0.18	44 (14.3)	45 (14.6)	0.04
Female (*n*, %)	11,247 (74.3)	150,593 (47.8)	0.61	11,247 (74.3)	11,243 (74.3)	<0.01
DM (*n*, %)	686 (4.5)	17,655 (5.6)	0.05	686 (4.5)	690 (4.6)	<0.01
HTN (*n*, %)	1657 (11)	26,579 (8.4)	0.08	1657 (11)	1703 (11.3)	<0.01
Dyslipidemia (*n*, %)	959 (6.3)	19,032 (6)	0.01	959 (6.3)	923 (6.1)	<0.01
Angina (*n*, %)	726 (4.8)	10,792 (3.4)	0.06	726 (4.8)	717 (4.7)	<0.01
AF (*n*, %)	9 (4.6)	3346 (1.1)	0.06	94 (0.6)	82 (0.5)	0.01
Heart disease (*n*, %)	518 (3.4)	9202 (2.9)	0.03	518 (3.4)	497 (3.3)	<0.01
Cerebrovascular disease (*n*, %)	1756 (11.6)	14,888 (4.7)	0.21	1756 (11.6)	1840 (12.2)	0.02
Chronic pulmonary disease (*n*, %)	1075 (7.1)	17,797 (5.6)	0.06	1075 (7.1)	1097 (7.3)	<0.01
Renal failure (*n*, %)	145 (1)	4355 (1.4)	0.04	145 (1)	142 (0.9)	<0.01
Chronic hepatitis (*n*, %)	583 (3.9)	17,223 (5.5)	0.08	583 (3.9)	559 (3.7)	<0.01
Anxiety (*n*, %)	641 (4.2)	3607 (1.1)	0.15	641 (4.2)	636 (4.2)	<0.01
Depression (*n*, %)	1874 (12.4)	7581 (2.4)	0.3	1874 (12.4)	1931 (12.8)	0.01
Sleep disorder (*n*, %)	912 (6)	5230 (1.7)	0.18	912 (6)	907 (6)	<0.01
Menopause (*n*, %)	584 (3.9)	7128 (2.3)	0.08	584 (3.9)	590 (3.9)	<0.01

PSM, propensity score matching; ASD, absolute standardized difference; SD, standard deviation; DM, diabetes mellitus; HTN, hypertension; AF, atrial fibrillation.

**Table 2 jpm-11-01331-t002:** Characteristics of the non-migraine headache and control groups before and after PSM.

	Before PSM			After PSM		
	NMH (*n* = 76,773)	Control (*n* = 315,330)	ASD	NMH (*n* = 14,606)	Control (*n* = 14,606)	ASD
Age, years (min, SD)	49.1 (14.6)	46.6 (15.4)	0.17	49 (14.6)	50 (14.9)	0.03
Female (*n*, %)	45,148 (58.8)	150,593 (47.8)	0.22	45,148 (58.8)	44,649 (58.2)	0.01
DM (*n*, %)	4471 (5.8)	17,655 (5.6)	<0.01	4471 (5.8)	4700 (6.1)	0.01
HTN (*n*, %)	9737 (12.7)	26,579 (8.4)	0.13	9737 (12.7)	10,361 (13.5)	0.02
Dyslipidemia (*n*, %)	5622 (7.3)	19,032 (6)	0.05	5622 (7.3)	5865 (7.6)	0.01
Angina (*n*, %)	4394 (5.7)	10,792 (3.4)	0.1	4394 (5.7)	4393 (5.7)	<0.01
AF (*n*, %)	911 (1.2)	3346 (1.1)	0.01	911 (1.2)	914 (1.2)	<0.01
Heart disease (*n*, %)	3014 (3.9)	9202 (2.9)	0.05	3014 (3.9)	3104 (4)	<0.01
Cerebrovascular disease (*n*, %)	9743 (12.7)	14,888 (4.7)	0.24	9743 (12.7)	9739 (12.7)	<0.01
Chronic pulmonary disease (*n*, %)	5871 (7.6)	17,797 (5.6)	0.08	5871 (7.6)	6085 (7.9)	0.01
Renal failure (*n*, %)	1070 (1.4)	4355 (1.4)	<0.01	1070 (1.4)	1095 (1.4)	<0.01
Chronic hepatitis (*n*, %)	3523 (4.6)	17,223 (5.5)	0.04	3523 (4.6)	3488 (4.5)	<0.01
Anxiety (*n*, %)	2679 (3.5)	3607 (1.1)	0.13	2679 (3.5)	2502 (3.3)	0.01
Depression (*n*, %)	4497 (5.9)	7581 (2.4)	0.15	4497 (5.9)	4636 (6)	<0.01
Sleep disorder (*n*, %)	2700 (3.5)	5230 (1.7)	0.1	2700 (3.5)	2782 (3.6)	<0.01
Menopause (*n*, %)	2602 (3.4)	7128 (2.3)	0.06	2602 (3.4)	2549 (3.3)	<0.01

NMH, non-migraine headache; PSM, propensity score matching; ASD, absolute standardized difference; SD, standard deviation; DM, diabetes mellitus; HTN, hypertension; AF, atrial fibrillation.

**Table 3 jpm-11-01331-t003:** ORs for vestibular and cochlear disorders in migraine patients after PSM.

	OR (95% CI)				
	MD	BPPV	VN	SNHL	Tinnitus
UA	2.519 (1.988–3.192)	1.941 (1.728–2.182)	2.838 (2.518–3.200)	1.647 (1.331–2.038)	1.867 (1.574–2.215)
AVA	2.597 (2.047–3.295)	2.045 (1.816–2.302)	2.976 (2.636–3.360)	1.740 (1.404–2.156)	1.972 (1.659–2.343)
AVPA	2.597 (2.047–3.295)	2.045 (1.816–2.302)	2.976 (2.636–3.360)	1.739 (1.404–2.155)	1.970 (1.658–2.341)

All *p*-values were *p* < 0.001. SPSS UA, unadjusted; AVA, all variables adjusted; AVPA, all variables plus propensity score adjusted; MD, Meniere’s disease; BPPV, benign paroxysmal positional vertigo; VN, vestibular neuronitis; SNHL, sensorineural hearing loss; OR, odds ratio; CI, confidence interval.

**Table 4 jpm-11-01331-t004:** ORs for vestibular and cochlear disorders in migraine subgroups after PSM.

	**ORs in the Migraine Group Taking Triptans (95% CI)**				
	**MD**	**BPPV**	**VN**	**SNHL**	**Tinnitus**
UA	2.781 (1.943–3.960)	2.222 (1.853–2.664)	2.683 (2.228–3.230)	1.749 (1.250–2.448)	1.732 (1.350–2.224)
AVA	2.930 (2.040–4.209)	2.365 (1.965–2.845)	2.883 (2.385–3.484)	1.821 (1.298–2.556)	1.859 (1.443–2.394)
AVPA	2.928 (2.038–4.206)	2.359 (1.960–2.838)	2.882 (2.385–3.483)	1.812 (1.291–2.543)	1.862 (1.446–2.399)
	**ORs in the Migraine Group Using NSAIDs (95% CI)**				
	**MD**	**BPPV**	**VN**	**SNHL**	**Tinnitus**
UA	1.746 (1.315–2.318)	2.314 (1.997–2.683)	3.190 (2.743–3.711)	1.568 (1.228–2.048)	1.841 (1.490–2.275)
AVA	1.809 (1.360–2.408)	2.434 (2.095–2.827)	3.364 (2.887–3.921)	1.649 (1.275–2.134)	1.984 (1.572–2.414)
AVPA	1.809 (1.360–2.408)	2.433 (2.095–2.826)	3.365 (2.887–3.921)	1.649 (1.274–2.133)	1.941 (1.566–2.405)

All *p*-values were *p* < 0.001. SPSS UA, unadjusted; AVA, all variables adjusted; AVPA, all variables plus propensity score adjusted; MD, Meniere’s disease; BPPV, benign paroxysmal positional vertigo; VN, vestibular neuronitis; SNHL, sensorineural hearing loss; NSAIDs, non-steroid anti-inflammatory drugs; OR, odds ratio; CI, confidence interval.

**Table 5 jpm-11-01331-t005:** ORs for vestibular and cochlear disorders in non-migraine headache patients after PSM.

	OR (95% CI)				
	MD	BPPV	VN	SNHL	Tinnitus
UA	1.749 (1.541–1.984)	1.681 (1.590–1.777)	1.990 (1.881–2.105)	1.339 (1.222–1.468)	1.640 (1.521–1.768)
AVA	1.774 (1.562–2.014)	1.731 (1.637–1.831)	2.048 (1.935–2.168)	1.395 (1.272–1.530)	1.693 (1.569–1.827)
AVPA	1.771 (1.560–2.011)	1.731 (1.637–1.831)	2.048 (1.935–2.168)	1.396 (1.273–1.531)	1.693 (1.569–1.826)

All *p*-values were *p* < 0.001. SPSS UA, unadjusted; AVA, all variables adjusted; AVPA, all variables plus propensity score adjusted; MD, Meniere’s disease; BPPV, benign paroxysmal positional vertigo; VN, vestibular neuronitis; SNHL, sensorineural hearing loss; OR, odds ratio; CI, confidence interval.

## Data Availability

Not applicable.

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
