# Peer review of "Risk of Vestibulocochlear Disorders in Patients with Migraine or Non-Migraine Headache"

_jpm, 2021, doi:10.3390/jpm11121331_

Round 1

Reviewer 1 Report

The authosa analyzed 10 years 17
of data from the Smart Clinical Data Warehouse. In patients with migraine and nMH, MD, BPPV, vestibular neuronitis (VN) and cochlear disorders, such as sensorineural hearing loss (SNHL) and tinnitus, were collected and compared to clinical data from controls who had health check-ups without headache.

The importance and validity of the proposed hypotheses are confirmed.

Good suitability and feasibility of the experimental and analysis methodology. the data are able to test the original proposed hypotheses.

Author Response

Dec 3, 2021

Reviewer 1

Journal of Personalized Medicine

Dear Reviewer 1,

Please find attached the revised version of our manuscript entitled “Risk of vestibulocochlear disorders in patients with migraine or non-migraine headache” (jpm-1473916).

We thank you for your thoughtful suggestions on your reading of the original version of our paper; most of the suggested changes have been incorporated into the revision.

All revisions are described in detail in the order mentioned in the review, following the reviewer’s critiques (in italics). We believe that the revisions have greatly improved the manuscript and hereby submit the revised version for consideration for publication.

Comments to author:

The authosa analyzed 10 years of data from the Smart Clinical Data Warehouse. In patients with migraine and nMH, MD, BPPV, vestibular neuronitis (VN) and cochlear disorders, such as sensorineural hearing loss (SNHL) and tinnitus, were collected and compared to clinical data from controls who had health check-ups without headache.

The importance and validity of the proposed hypotheses are confirmed.

Good suitability and feasibility of the experimental and analysis methodology. the data are able to test the original proposed hypotheses.

We appreciate the reviewer for these comments.

We have tried to address the issues raised by the reviewers and editorial board member. We are grateful for the constructive comments that arose during the review process. We believe that our paper has been improved based on these suggestions.

Yours faithfully,

Jong-Hee Sohn, M.D. Ph.D.

Department of Neurology, Chuncheon Sacred Heart Hospital, Hallym University College of Medicine, 77 Sakju-ro, Chuncheon-si, Gangwon-do, 24253, Republic of Korea

Tel: +82-33-252-9970, Fax: +82-33-241-8063

Reviewer 2 Report

The major concern in reaching conclusions from such data is the heavy reliance on the accuracy of coded diagnoses. For example, is it possible that some cases of "tinnitus" were describing phonophobia? What criteria were used for diagnosing MD? Could some of these vestibular or cochlear symptoms be manifestations of VM rather than an independent vestibular or cochlear disorder. Distinction between VM and vestibulo-cochlear diagnoses may be very difficult in individual cases so the confidence that all such cases would be coded correctly is diminished.

Nevertheless, this is stimulating work which could perhaps be supported further by in-depth examination and analysis of the diagnostic accuracy in at least a sample of cases. 

Some discussion of these points is warranted.

Author Response

Dec 3, 2021

Reviewer 2

Journal of Personalized Medicine

Dear Reviewer 2,

Please find attached the revised version of our manuscript entitled “Risk of vestibulocochlear disorders in patients with migraine or non-migraine headache” (jpm-1473916).

We thank you for your thoughtful suggestions on your reading of the original version of our paper; most of the suggested changes have been incorporated into the revision.

All revisions are described in detail in the order mentioned in the review, following the reviewer’s critiques (in italics). We believe that the revisions have greatly improved the manuscript and hereby submit the revised version for consideration for publication.

Comments to author:

The major concern in reaching conclusions from such data is the heavy reliance on the accuracy of coded diagnoses. For example, is it possible that some cases of "tinnitus" were describing phonophobia? What criteria were used for diagnosing MD? Could some of these vestibular or cochlear symptoms be manifestations of VM rather than an independent vestibular or cochlear disorder. Distinction between VM and vestibulo-cochlear diagnoses may be very difficult in individual cases so the confidence that all such cases would be coded correctly is diminished. Nevertheless, this is stimulating work which could perhaps be supported further by in-depth examination and analysis of the diagnostic accuracy in at least a sample of cases.

Some discussion of these points is warranted.

We thank the reviewer for these comments and specific suggestions, which have improved our manuscript.

We agree with this important comment.

Our study design was retrospective, and unfortunately, we did not collect exact clinical information regarding vestibulocochlear disorders or headache characteristics. The symptoms of VM may mimic to the those of MD and BPPV. In particular, the primary differential diagnosis is MD. The current diagnostic criteria for both of these disorders are mainly based on the clinical symptoms of a patient because no biological markers are available. Also, the signs and symptoms of VM sometimes overlap with those of MD, which may give rise to diagnostic uncertainty. In the early stages of these disorders, a differential diagnosis between MD and VM may be difficult. Due to the similar clinical presentations of these disorders, a follow-up assessment is the only method that can accurately distinguish between VM and MD because most reliable distinguishing feature of patients with MD are progressive hearing loss that manifests over several years and low-frequency hearing loss evident on an audiogram. Therefore, the diagnosis of vestibulocochlear disorders may be difficult and uncertain, which can reduce the accuracy of diagnostic coding in these cases. Additional longitudinal follow-up clinical investigation are needed to evaluate the accuracy of diagnosing the vestibulocochlear disorders including detailed clinical features and laboratory tests for vestibulocochlear function.

As recommended, we have revised and added several sentences to the Discussion, as follows:

In addition, detailed clinical information regarding headache characteristics or vestibulo-cochlear disorders was not collected. The symptoms of VM may mimic to the those of MD and BPPV (47). In particular, the primary differential diagnosis is MD. The current diagnostic criteria for both of these disorders are mainly based on the clinical symptoms of a patient because no biological markers are available. Also, the signs and symptoms of VM sometimes overlap with those of MD, which may give rise to diagnostic uncertainty (10-14, 48). In the early stages of these disorders, a differential diagnosis between MD and VM may be difficult. Due to the similar clinical presentations of these disorders, a follow-up assessment is the only method that can accurately distinguish between VM and MD because most reliable distinguishing feature of patients with MD are progressive hearing loss that manifests over several years and low-frequency hearing loss evident on an audiogram (49). Therefore, the diagnosis of vestibulocochlear disorders may be difficult and uncertain, which can reduce the accuracy of diagnostic coding in these cases. To evaluate the accuracy of diagnosing the vestibulocochlear disorders, longitudinal follow-up clinical investigation, including detailed clinical features and laboratory test for vestibulocochlear function are required.

(page 9, lines 308 – page 9, lines 322)

We have also added citations:

  1. Neff BA, Staab JP, Eggers SD, Carlson ML, Schmitt WR, Van Abel KM, et al. Auditory and vestibular symptoms and chronic subjective dizziness in patients with Meniere's disease, vestibular migraine, and Meniere's disease with concomitant vestibular migraine. Otol Neurotol (2012) 33(7):1235–44. doi: 10.1097/MAO.0b013e31825d644a. PubMed PMID:22801040.

  1. Lopez-Escamez, Dlugaiczyk J, Jacobs J, Lempert T, Teggi R, von Brevern M, et al. Accompanying symptoms overlap during attacks in Meniere's Disease and vestibular migraine. Front in Neurol (2014) 15(5):265. doi: 10.3389/fneur.2014.00265. PubMed PMID:25566172.

  1. Radtke A, Lempert T, Gresty MA, Brookes GB, Bronstein AM, Neuhauser H. Migraine and Meniere's disease: Is there a link? Neurology (2002) 10;59(11):1700–4. doi: 10.1212/01.wnl.0000036903.22461.39. PubMed PMID: 12473755.

We have tried to address the issues raised by the reviewers and editorial board member. We are grateful for the constructive comments that arose during the review process. We believe that our paper has been improved based on these suggestions.

Yours faithfully,

Jong-Hee Sohn, M.D. Ph.D.

Department of Neurology, Chuncheon Sacred Heart Hospital, Hallym University College of Medicine, 77 Sakju-ro, Chuncheon-si, Gangwon-do, 24253, Republic of Korea

Tel: +82-33-252-9970, Fax: +82-33-241-8063
